# Implementation of e-mental health interventions for informal caregivers of adults with chronic diseases: a protocol for a mixed-methods systematic review with a qualitative comparative analysis

Chelsea Coumoundouros [1], Louise von Essen,[1] Robbert Sanderman,[2,3] Joanne Woodford[1]

For numbered affiliations see end of article.

**Correspondence to**
Ms Chelsea Coumoundouros; chelsea.coumoundouros@kbh.uu.se

## ABSTRACT

**Introduction** Informal caregivers provide the majority of care to individuals with chronic health conditions, benefiting the care recipient and reducing use of formal care services. However, providing informal care negatively impacts the mental health of many caregivers. E-mental health interventions have emerged as a way to provide accessible mental healthcare to caregivers. Much attention has been given to reviewing the effectiveness and efficacy of such interventions, however, factors related to implementation have received less consideration. Therefore, this mixed-methods systematic review will aim to examine factors associated with the effectiveness and implementation of e-mental health interventions for caregivers.

**Methods and analysis** Eligible studies published since 1 January 2007 will be searched for in several electronic databases (CINAHL Plus with Full Text, the Cochrane Library, EMBASE, PsycINFO, PubMed and Web of Science), clinical trial registries and OpenGrey, with all screening steps conducted by two independent reviewers. Studies will be included if they focus on the implementation or effectiveness of e-mental health interventions designed for informal adult caregivers of adults with cancer, heart disease, stroke, diabetes, dementia or chronic obstructive pulmonary disease. Pragmatic randomised controlled trials quantitatively reporting on caregiver anxiety, depression, psychological distress or stress will be used for a qualitative comparative analysis to identify combinations of conditions that result in effective interventions. Qualitative and quantitative data on implementation of e-mental health interventions for caregivers will be integrated in a thematic synthesis to identify barriers and facilitators to implementation. These results will inform future development and implementation planning of e-mental health interventions for caregivers.

**Ethics and dissemination** Ethical approval is not required for this study as no primary data will be collected. Results will be disseminated in the form of a scientific publication and presentations at academic conferences and plain language summaries for various stakeholders.

**PROSPERO registration number** CRD42020155727.

### Strengths and limitations of this study

► The mixed-method design of this review will ensure a wide variety of data on implementation is captured and interpretations account for both qualitative and quantitative research findings.

► The peer-reviewed, comprehensive search strategy with all selection steps completed by two independent reviewers will ensure a thorough search of the literature and reduce bias in study selection.

► High heterogeneity across studies in terms of implementation or intervention features is easily accommodated in a qualitative comparative analysis.

► Crisp set qualitative comparative analysis produces concrete results, increasing the usability of findings for healthcare professionals and decision-makers.

► However, crisp set qualitative comparative analysis dichotomises all variables including the outcome, therefore, a more detailed understanding of the strength of the effect size may be lost in this process.

## INTRODUCTION

Informal caregivers (hereafter referred to as caregivers) are family members or friends who provide unpaid support and care to individuals with healthcare needs. Caregivers play a vital societal role in healthcare systems worldwide, providing up to 80% of care to individuals with long-term care needs.[1] Informal care provision can include emotional support, assistance with household tasks (eg, cooking, cleaning), medical care, transportation, managing finances and advocacy on behalf of the care recipient.[2,3] Demand for caregivers is expected to increase in the future as the proportion of older adults in populations around the world increases and healthcare policies favour deinstitutionalisation and outpatient care.[4,5] As societal dependence on informal care continues to

BMJ

grow, it is becoming increasingly important to implement programmes and policies to support individuals who become caregivers.[2 6]

Caregivers can experience both positive and negative outcomes over the course of their time as a caregiver.[3 7 8] Caregiving can lead to an improved relationship between caregiver and care recipient, feelings of personal development and a sense of accomplishment related to obtaining skills and recognising the impact of the care they provide.[8] However, caregivers may also experience negative outcomes related to the caregiving role, such as financial strain and poor physical and mental health.[3 7 9] Indeed, the rate of depression and anxiety among caregivers exceeds that of the general population.[10] The prevalence of depressive symptoms in cancer and stroke caregivers is often above 40% and the prevalence of anxiety ranges from 21% to over 40%.[11 12] Mental health problems can result in large personal and societal costs related to increased morbidity and reduced productivity.[13–15] Additionally, poor caregiver mental health negatively impacts distress levels in the care recipient[16 17] and the quality of care provided by the caregiver.[18–21]

There is a clear need to develop effective interventions and resources to prevent or reduce the mental health burden experienced by caregivers. However, caregivers have reported various barriers to accessing mental health services such as lack of knowledge regarding available services, financial barriers, stigma and prioritisation of the caregiving role over self-care.[22] E-health technologies have emerged as an accessible way to provide support and information to caregivers[23–25] and can be designed to achieve various goals such as, improving communication, teaching skills or reducing depression.[26 27] Numerous systematic reviews and meta-analyses have examined e-health interventions for caregivers of adults with chronic health conditions, showing the potential for e-health interventions to improve caregiver well-being.[23 25 28–40]

E-mental health interventions, that is, mental health interventions delivered via the internet or using mobile technologies,[27 41 42] represent a subset of e-health interventions. E-mental health solutions offer a means to improve mental health service access globally[43 44] by eliminating many barriers to mental health service access (eg, transportation, stigma, time)[25 42 45] and are often more cost-effective than traditional therapies.[42 45] Meta-analyses show reductions in caregiver's depression and anxiety in response to e-mental health interventions.[37 39] However, as many reviews focus on intervention efficacy and effectiveness,[23 28–32 34 36–40 46] gaps remain in our understanding of factors related to the intervention and the implementation context that make e-mental health interventions effective among caregivers.

Wider literature suggests that the implementation of e-health programmes in real-world settings often encounters many barriers,[24 33 47 48] preventing effective interventions from being made available to those who need them. Few reviews have examined factors related to implementation of e-health interventions for caregivers,[33 35] with no current reviews, to the best of our knowledge, focusing on implementation of e-mental health interventions for caregivers exclusively. Evaluating the implementation of an intervention is essential to gain insights into why interventions succeed or fail when put into practice. Factors influencing implementation can be related to the intervention itself, the participants, the implementation setting and wider societal factors (eg, regional policies).[49] Trials with a more pragmatic design may be better suited to investigating factors potentially associated with implementation given real-world conditions are more closely reflected in pragmatic trials.[50 51] However, systematic reviews and meta-analyses do not often distinguish between pragmatic and explanatory (also referred to as efficacy) trials despite the different conditions (eg, setting, recruitment methods, eligibility criteria, control of adherence to and delivery of the intervention) under which interventions are evaluated.[50 52] Identifying trials with a pragmatic design may be a valuable factor to consider when interpreting results of reviews to inform implementation.

The aim of this review is to examine factors related to the effectiveness and implementation of e-mental health interventions for caregivers of adults with chronic diseases. Two approaches will be used to investigate this. First, studies with more pragmatic designs will be used exclusively to determine which combinations of intervention or implementation characteristics are associated with effectiveness using a qualitative comparative analysis. Second, reports regarding the implementation of e-mental health interventions will be thematically synthesised to establish the common barriers and facilitators to e-mental health implementation. Findings from this review can be used to guide the development of effective e-mental health interventions to support caregivers and ensure the successful implementation of these interventions within real-world healthcare settings.

## METHODS AND ANALYSIS

The Preferred Reporting Items for Systematic Reviews and Meta-Analyses Protocols (PRISMA) checklist[53] (online supplementary appendix 1) and the Joanna Briggs methodology for mixed-methods systematic reviews[54] were used to guide the development of this protocol. Any protocol amendments will be recorded in PROSPERO.

### Study eligibility criteria

The eligibility criteria used to inform study inclusion and exclusion are outlined using population, interventions, comparators, outcomes and study design (PICOS).[55 56]

#### Population

Unpaid adult caregivers (aged 18 years or older) of adults with either cancer, chronic obstructive pulmonary disease (COPD), dementia, diabetes, heart disease or stroke. Care recipient's chronic health conditions eligible for inclusion were selected as, globally, they are the largest

sources of disability adjusted life years due to physical chronic diseases in adults[57] and often require informal care.[58] No restrictions will be placed on the frequency or amount of care provided for someone to be considered a caregiver. Studies exclusively focusing on caregivers with severe mental health conditions (eg, psychosis or bipolar disorder) will be excluded, as the focus of this review is on e-mental health interventions targeting psychological health difficulties associated with the provision of informal care, for example anxiety or depression, as opposed to targeting severe mental health conditions. Studies with interventions that solely focus on caregivers providing care to non-community dwelling care recipients will be excluded, given caregivers of individuals who do not live in the community may spend less time providing informal care[59] and generally experience lower levels of depression.[60 61] Additionally, studies of interventions designed specifically for caregivers of individuals at the end-of-life (eg, within a few months of death) will be excluded, as end-of-life caregiving is associated with additional needs and burdens, for example difficulties related to grief and bereavement.[62]

### Interventions

Interventions will use internet technology, such as web-based platforms or mobile-based applications, to deliver a mental health intervention to caregivers.[27 41] E-mental health interventions can encompass many types of mental health support such as screening, prevention, treatment or service delivery.[41] This review will focus on interventions targeting the treatment of common caregiver psychological health difficulties (anxiety, depression, psychological distress or stress). This can include any type of mental health treatment, including psychoeducation. Psychoeducation is defined as the provision of information regarding common psychological health difficulties and can be delivered passively (eg, an information website) or actively (eg, an information website with therapist support, homework or exercises).[63] The majority of therapeutic materials within the e-mental health intervention must be internet based, however, this may be supplemented with additional forms of support (such as telephone contact, face-to-face support or video conferencing). There are no restrictions on the amount of support provided within the e-mental health intervention. Interventions delivered via telephone, CD-ROM (Compact Disc-Read Only Memory) or video (including Skype) alone will be excluded.

### Comparators

As it is necessary to determine effect sizes for the qualitative comparative analysis,[64] only studies of pragmatic randomised controlled trials with non-active controls will be included in this analysis. Non-active controls include: no treatment, wait-list control, treatment as usual, non-specific treatment component control (eg, control for attention) or education on the care recipient's condition.[65] Studies using psychoeducation or active controls

(eg, controls using specific treatment components or studies comparing two therapies) will be excluded.

For thematic synthesis of barriers and facilitators to implementation, studies of any design (eg, randomised controlled trials, process evaluations, focus groups) will be included in the analysis, regardless of the presence or absence of a control.

### Outcomes

For the qualitative comparative analysis, studies must report on caregiver mental health outcomes, specifically anxiety, depression, psychological distress or stress, measured using an instrument with at least acceptable reliability (Cronbach's alpha ≥0.7).[66] Reliability of outcome measures will be assessed based on the main validation paper of the relevant measurement instrument, as this review will likely include studies with different caregiver populations, ages, genders and languages, the combination of which may not have been validated. Examples of eligible measurement instruments include the Centre for Epidemiologic Studies-Depression Scale,[67] the Hospital Anxiety and Depression Scale[68] or the Perceived Stress Scale.[69]

For the thematic synthesis, studies will report on barriers and/or facilitators to intervention implementation. This may include qualitative (eg, interviews or focus groups) or quantitative (eg, Normalisation Measure Development questionnaire[70]) data. Barriers or facilitators can include factors related to any aspect of the Consolidated Framework for Implementation Research[49] or the implementation outcome framework developed by Proctor *et al*.[71] The Consolidated Framework for Implementation Research consists of five domains related to implementation, namely (1) intervention characteristics (eg, adaptability, complexity); (2) outer setting (eg, external policies, patient needs and resources); (3) inner/implementation setting (eg, culture within the organisation, readiness for implementation); (4) characteristics of individuals (eg, self-efficacy, individual stage of change) and (5) process (eg, planning, engaging).[49] The implementation outcome framework broadly classifies measurable implementation outcomes which includes acceptability, adoption, feasibility, fidelity, reach, appropriateness, implementation cost and sustainability.[71]

### Study designs

Studies included for the qualitative comparative analysis must be pragmatic randomised controlled trials (also referred to as effectiveness trials). Pragmatic trials will be identified using the validated PRagmatic Explanatory Continuum Indicator Summary 2 (PRECIS-2) tool.[50] PRECIS-2 was developed with input from clinicians, researchers and policy-makers to allow trialists to assess how pragmatic or explanatory their trial design is across nine domains: eligibility criteria, recruitment, setting, organisation, flexibility (delivery), flexibility (adherence), follow-up, primary outcome and primary analysis.[50] Trials with a pragmatic design will be defined as any trial with

a mean score of 3 or higher using the PRECIS-2 tool.[50] PRECIS-2 has been used with this cut-off score to categorise studies in another systematic review,[72] although to our knowledge it has not previously been used to exclude studies from a systematic review. Using a cut-off score of 3 should ensure generous inclusion of trials containing at least a mixture of pragmatic and explanatory design features.[50]

To assess barriers and facilitators to implementation, any study type with quantitative and/or qualitative data will be eligible for inclusion.

### Search strategy

Comprehensive literature searches will be conducted in multiple electronic databases (CINAHL Plus with Full Text, the Cochrane Library, EMBASE, PsycINFO, PubMed and Web of Science). Clinical trial registries ( www.clinicaltrials.gov and www.who.int/trialsearch/) will be searched for relevant completed clinical trials and the resulting publications will be found and screened for inclusion. Searches for grey literature will be performed using OpenGrey (http://www.opengrey.eu/), a database of grey literature in Europe such as research reports and conference papers.

The search strategy was developed in consultation with Agnes Kotka, a librarian at Uppsala University and was reviewed by Professor Mariët Hagedoorn and Truus van Ittersum (University Medical Centre Groningen, University of Groningen) and Dr. Nathan Davies (University College London) following the PRESS peer review guidelines[73] (online supplementary appendix 2). The search was constructed using terms related to (1) caregivers; (2) the chronic health conditions of interest (cancer, COPD, dementia, diabetes, heart disease and stroke); (3) e-health/information and communication technology; (4) mental health and (5) psychological therapies (see online supplementary appendix 3). Included terms were informed by existing reviews focusing on the population and/or intervention of interest to this review.[9 33 39 74–79] Search terms were refined based on feedback from the peer-review process, resulting in the addition of more truncations to search terms, elimination of repetitive search terms that did not retrieve additional records and the addition of an abbreviation missed prior to the peer-review process. The search will include relevant Medical Subject Headings when possible and terms will be searched for in the title/abstract of publications. Included studies will be restricted to those published in English, Dutch, German or Swedish. Literature produced from January 2007 onwards will be eligible for inclusion. Technologies from work published prior to 2007 may be outdated and other reviews have shown that production of publications involving e-health began to rise from 2007 onwards.[33 35] Electronic searches will be rerun prior to reporting of results to ensure the search is as up to date as possible.

On final inclusion of any studies, their references, results from forward citation searches and from the first three pages of the 'find similar' search function in PubMed will be used to check for any additional studies of interest. Experts in the field will be contacted to identify further studies for inclusion.

### Study selection

Results of database searches will be imported into EndNote for deduplication following the procedures outlined by Bramer *et al*.[80] Remaining records will be imported into the online screening software Rayyan.[81] Titles, abstracts and full texts will be screened independently by two reviewers. Conflicts will be discussed and a third reviewer will be consulted if consensus cannot be reached. Study selection will be based on the criteria outlined by the PICOS, with reasons for study exclusion being recorded at the full-text screening stage. Full texts will be checked against each sub-section of the PICOS, recording which sub-sections are or are not met by each study, with an overall reason for exclusion being reported in the PRISMA flow diagram. This will facilitate detailed discussions regarding study exclusion when conflicts arise. If studies do not contain enough information to decide on inclusion, the original authors will be contacted at most twice over a 1-month period to obtain information to determine study eligibility. If the original authors do not respond, the study will be excluded. Abstracts, theses, books, commentaries, editorials and letters to the editor will be excluded. Reviews and study protocols will also be excluded, however, the references of related reviews will be checked for additional studies of interest, published results of relevant study protocols will be obtained and if protocol results are unpublished, authors will be contacted to determine whether access to unpublished results is possible.

Records retrieved from searches of clinical trial registries and OpenGrey will be screened for eligibility by one reviewer. When relevant clinical trial registries are identified, any resulting publications will be retrieved and screened for inclusion, unless already captured by the electronic database searches. If results from relevant trial registries are unpublished, authors will be contacted to determine if they are able to share details of any available results. Authors of grey literature records that do not contain enough information to assess eligibility will also be contacted for additional study details.

Exclusion of studies on the basis of adopting a more explanatory, as opposed to pragmatic, trial design will be conducted as a final step during the full-text screening process. This screening step will only be applied to trials eligible for the qualitative comparative analysis. Studies will be scored using the PRECIS-2 tool by two independent reviewers and studies with a mean score below 3 will be excluded.[50 72]

### Assessment of methodological quality

Methodological quality of studies included in the qualitative comparative analysis will be evaluated using the Cochrane Risk of Bias 2.0 tool for randomised controlled

trials.[82] [83] This evaluation will facilitate the identification of selection, performance, measurement, attrition and reporting bias.[83] Authors will be contacted if more information is required to complete the quality assessment. Reporting bias will be explored by comparing outcomes measures described in study protocols to the outcome measures reported in the methods and results sections of the corresponding completed trial. In response to any identified inconsistencies, authors will be contacted to determine potential causes of this. Study assessment will be conducted by two independent reviewers, followed by discussion of any discrepancies, consulting a third reviewer as needed. Studies will not be excluded based on methodological quality, however the results of the Cochrane Risk of Bias 2.0 evaluation will be reported descriptively.

### Data extraction

Data from included full texts will be extracted into Microsoft Excel (2016), using a data extraction form developed for this review based on the Centre for Reviews and Dissemination guidelines.[56] Extracted information will include data pertaining to study participants, study design, the intervention and relevant outcomes (full details in online supplementary appendix 4). Data used in the qualitative comparative analysis and thematic synthesis will be extracted independently by two reviewers, with resulting extractions compared for accuracy and completion. All other data will be extracted by one reviewer and verified by a second reviewer. If conflicts arise, the original publication will be referred to in order to resolve misunderstandings and a third reviewer will be consulted if necessary. Authors will be contacted at most twice to obtain additional data and/or clarification as needed. Qualitative results pertaining to implementation will be transferred into NVivo V.10 software[84] for thematic synthesis.

### Data synthesis

Data related to the characteristics of each included study, such as the sample (eg, sample size, participant demographics) or intervention (eg, duration, type of support provided, delivery mode) characteristics, will be reported in summary tables. Further data synthesis will involve two analysis methods. Pragmatic randomised controlled trials with quantitative mental health outcome data will be included in the qualitative comparative analysis. Publications of any study design reporting on implementation will be included in the thematic synthesis, taking an integrative approach to synthesise both qualitative and quantitative findings.

#### Qualitative comparative analysis

A crisp set qualitative comparative analysis will be conducted to determine sets of conditions that result in effective e-mental health interventions for caregivers.[64] Crisp set qualitative comparative analysis involves dichotomising outcome data (eg, effective or not effective) and conditions (eg, present or absent) selected for inclusion in the analysis into distinct categories.[64] A crisp set analysis approach was selected over a fuzzy set analysis as the results will be more clearly interpretable and easier for decision-makers to use.[85]

The first step of a qualitative comparative analysis is to build a data table containing information regarding the effectiveness of each study and conditions related to the intervention and its implementation (see online supplementary appendix 5).[64] Conditions to include in the data table will be based on important factors related to intervention components (eg, uses goal-setting, homework), intervention delivery methods (eg, mobile app, computer) and implementation (eg, acceptability, feasibility). By restricting this analysis to pragmatic trials, which are designed to more closely reflect real-world settings, implementation conditions are more likely to be reported. Conditions selected will be adjusted given the need to ensure adequate heterogeneity is present.[64] Qualitative comparative analysis requires diversity among studies in terms of conditions present and intervention effectiveness in order to determine the combination of factors sufficient for interventions to be effective.[64] Therefore, adjustments to outcome classification and conditions selected for analysis will be needed after data collection is completed.

Intervention effectiveness will be measured as the standardised mean effect size between control and comparator groups' mental health outcomes, calculated using Hedges' g and the Comprehensive Meta-Analysis (V.3) software. Effect sizes will be calculated for all mental health outcomes of interest for this review (anxiety, depression, psychological distress and stress) and will be based on data collected immediately after intervention completion. If enough studies report subsequent postintervention follow-ups, these effect sizes will be calculated to explore whether different factors contribute to sustained intervention success. Effect sizes will be used to create crisp sets to categorise studies as effective (Hedges' $g \geq 0.3$) or not effective.[86] If most interventions are effective (or not effective), a different classification system will be created to ensure adequate heterogeneity for analysis,[64] for example, categorising studies as highly effective (Hedges' $g \geq 0.5$) or not highly effective.[86] Proposed cutoffs were developed based on existing meta-analyses of e-mental health interventions.[87–90]

The main data table will use general effectiveness as the outcome measure, meaning the primary mental health outcome as identified in each study will be used to represent the effectiveness of that intervention. If studies include multiple outcomes of interest, but do not identify a primary outcome measure, the outcome most frequently measured in included studies will be used to evaluate intervention effectiveness. Secondary analyses may be conducted for anxiety, depression, psychological distress and stress separately, to explore whether different conditions are more important for different outcome measures. However, this is dependent on identification of an adequate number of studies for each outcome of

interest. After completion of the data table, truth tables will be constructed and the software fs/QCA (V.3.1b) will be used to determine the sufficient conditions for effective e-mental health interventions.[91 92]

## Thematic synthesis

Data from studies addressing implementation of e-mental health interventions for caregivers will be thematically synthesised using a deductive coding approach, to identify barriers and facilitators experienced during implementation.[93 94] It will likely be necessary to integrate qualitative and quantitative data as many aspects of implementation such as acceptability, feasibility and usability, may be measured using quantitative tools.[95] First, qualitative data will be thematically analysed using the Consolidated Framework for Implementation Research to guide coding.[49] This framework was selected a priori as it was developed by combining multiple implementation theories into a single, comprehensive theory covering all aspects related to implementation[49] and it has been used as a coding guide in other reviews on implementation.[33 96] Qualitative data will be coded based on the 39 pre-defined constructs within the Consolidated Framework for Implementation Research,[49] with the creation of additional codes if needed.

Quantitative data will be narratively summarised to facilitate subsequent integration of qualitative and quantitative findings. Creating narrative summaries will involve approaches such as textually describing study findings and grouping findings based on the constructs and domains of the Consolidated Framework for Implementation Research.[94] Initially, 10% of full texts included in the thematic synthesis will be coded independently by two reviewers, followed by discussion of the coding process in consultation with a third reviewer. The remaining coding will be conducted by one reviewer with regular discussions with a second reviewer, involving a third reviewer as needed. Results of the initial coding of qualitative data and narrative summaries of quantitative data will be analysed together to identify barriers and facilitators to implementation. Two reviewers will independently identify barriers and facilitators, followed by discussion involving a third reviewer as needed.[93] Through this discussion, more abstract, analytical themes will be developed that go beyond the initial codes and identified barriers and facilitators.[93] This process will be iterative, modifying barriers and facilitators after defining initial analytical themes, followed by further refinement of analytical themes until the analytical themes fully encompass all codes and identified barriers and facilitators.[93]

## Patient and public involvement

There was no patient or public involvement in the development of this protocol. However, caregivers and healthcare professionals will be involved in the interpretation of data for the thematic synthesis.[97 98] Results of the thematic synthesis will be presented to a multidisciplinary group of stakeholders (eg, caregivers, mental health service providers) to explore whether the identified themes resonate with their experiences or if they feel important aspects related to implementation have not been captured by the synthesis. Their perspectives will be incorporated into the interpretation of the results of the thematic synthesis.

## DISCUSSION

Despite the importance of implementation planning and need for effective e-mental health interventions for caregivers in real world practice, there have been no reviews focusing on this area. Using pragmatic trials and implementation research, this review will identify both the key characteristics of effective interventions and barriers and facilitators to implementation. A qualitative comparative analysis will be employed to identify combinations of conditions resulting in effective e-mental health interventions for caregivers, a method which, to the best of our knowledge, has not yet been used in this field. The results of the qualitative comparative analysis can be used to improve the design of future e-mental health interventions by identifying intervention components and implementation factors important to intervention effectiveness in real-world settings.

Additionally, common barriers and facilitators to implementation of e-mental health interventions for caregivers identified in this review can be used to inform implementation planning for similar interventions designed to reduce the mental health burden experienced by caregivers. For example, results may highlight the importance of providing training to individuals delivering the intervention or involvement of management staff in implementation activities. Improving our understanding of factors associated with implementation will allow implementers to both account for and avoid common implementation challenges, thereby potentially increasing subsequent uptake and effectiveness of e-mental health programmes developed to support caregivers.

## Ethics and dissemination

The results of this work will be disseminated in the form of a scientific publication in a peer-reviewed journal and as presentations at conferences. Plain language summaries will be prepared and provided to groups working with or supporting caregivers and healthcare organisations. Results will also be disseminated throughout the Marie Sklodowska-Curie Innovation Training Network, ENTWINE, which conducts research related to informal care and technological interventions to support caregivers.

**Author affiliations**
[1]Clinical Psychology in Healthcare, Department of Women's and Children's Health, Uppsala University, Uppsala, Sweden
[2]Department of Health Psychology, University of Groningen, University Medical Center Groningen, Groningen, The Netherlands
[3]Department of Psychology, Health and Technology, University of Twente, Enschede, The Netherlands

**Acknowledgements** We would like to thank Agnes Kotka, librarian at Uppsala University, for assisting in development of the search strategy. We are also grateful to Professor Mariët Hagedoorn, Truus van Ittersum (University Medical Centre Groningen, University of Groningen) and Dr. Nathan Davies (University College London), for providing valuable feedback as the peer-reviewers of the search strategy.

**Contributors** CC contributed to the design of the study and wrote the manuscript. JW, who acts as the reviews guarantor, conceived the study, contributed to the study design and critically revised the manuscript draft. LvE and RS critically revised the study design and manuscript. All authors approved of the final manuscript.

**Funding** This work was supported by the European Union's Horizon 2020 research and innovation programme under the Marie-Sklodowska Curie grant agreement no 814 072.

**Disclaimer** Funders were not involved in the creation, development or publication of this protocol, nor will they be involved in the conduct, analysis or reporting of the resulting systematic review.

**Competing interests** None declared.

**Patient consent for publication** Not required.

**Provenance and peer review** Not commissioned; externally peer reviewed.

**ORCID iD**
Chelsea Coumoundouros http://orcid.org/0000-0001-5539-974X

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
