## [Reviewer comments · BMJ Open]

ARTICLE DETAILS

TITLE (PROVISIONAL)	Implementation of e-mental health interventions for informal caregivers of adults with chronic diseases: a protocol for a mixed methods systematic review with a qualitative comparative analysis
AUTHORS	Coumoundouros, Chelsea; von Essen, Louise; Sanderman, Robbert; Woodford, Joanne

VERSION 1 – REVIEW

REVIEWER	Dr Charlene Treanor Centre for Public Health, Queen's University Belfast, United Kingdom
REVIEW RETURNED	27-Nov-2019

GENERAL COMMENTS	This is a very well-written protocol for a mixed-methods systematic review addressing implementation of e-mental health interventions for informal caregivers of adults with chronic illness. The authors present a very comprehensive plan for undertaking this review and I wish them luck with this task. I have a few minor suggestions for the authors to consider. 1. Please provide a rationale for the year restriction on your search reported in the abstract, but not in the methods section.2. 'Study eligibility criteria': Please provide a rationale for why the excluded caregiver populations were excluded and define how psycho-education differs from education.3. 'Outcomes': Please clarify if the reliability of outcome measures will be based on caregiver populations or a general population as outcome measures may perform differently in different populations.4. 'Search Strategy': Please report what steps will be taken when identifying registered trial and other grey literature.
--

REVIEWER	Miriam C. Noonan University of Exeter, United Kingdom
REVIEW RETURNED	14-Feb-2020

GENERAL COMMENTS	I enjoyed reading this protocol and I thought the intro/background was very well written and presented the topic very well. It is an interesting topic and I look forward to reading the outcomes of the review. I would just suggest some minor revisions. Pragmatic RCT - please explain this much earlier in the text
---

	I found it confusing to grasp between RCTs for your QCA analysis and the fact that you are examining any study for your Thematic synthesis. Can this be made clearer earlier. e.g. you use the word reports at one point to describe the studies you would be looking at for your Thematic synthesis and this was confusing as I wasn't sure what type of reports this would be. Include rationale for why you are excluding studies with caregivers with mental health issues, non-community living patients and palliative care. Interventions (PICO) - Can you clarify whether RCTs, observational, cohort etc. Thematic synthesis - needs a bit more clarity on how this will be conducted. It reads more that it is a framework analysis and you will be generating themes. More explanation on how you plan to integrate the data sets and how you will take into account Quant. elements that may be lost in synthesis e.g. sample size, attrition, duration of intervention etc.
--	--

REVIEWER	Heejung Kim Yonsei University, College of Nursing, Republic of Korea
REVIEW RETURNED	21-Feb-2020

GENERAL COMMENTS	Thank you for giving me an opportunity to review this protocol. I expect that this paper will be very helpful for caregiving research and practitioners. However, there are few concern you need to consider. I hope that my review helps you. 1. This review will be very heterogeneous because it included many types of disease, both qualitative and quantitative methods, and diverse caregiver's psychological outcomes. So, it is very concerned that it will be difficult to synthesize the findings to draw key conclusion. Thus, you may think about narrowing down the scope of diseases and caregiver's outcomes. 2. In contrast, you set up too narrow scope for the eHealth. You seem to include only mobile- internet based. However, recently, there are many types along with smart devices. I suggest that you have consultation with ITC experts about this issue. Other than that, this review will be very interesting and helpful for us. Good luck!
--

VERSION 1 – AUTHOR RESPONSE

Reviewer(s)' Comments to Author:

Reviewer: 1
Reviewer Name
Dr Charlene Treanor

Institution and Country
Centre for Public Health, Queen's University Belfast, United Kingdom

Please state any competing interests or state 'None declared':
None declared

Please leave your comments for the authors below: This is a very well-written protocol for a mixed-methods systematic review addressing implementation of e-mental health interventions for informal caregivers of adults with chronic illness. The authors present a very comprehensive plan for undertaking this review and I wish them luck with this task. I have a few minor suggestions for the authors to consider.

1. Please provide a rationale for the year restriction on your search reported in the abstract, but not in the methods section.

Response: The year restriction and rationale is reported in the methods section titled “Search strategy”, page 7 line 34-37 as follows:

“Literature produced from January 2007 onwards will be eligible for inclusion. Technologies from work published prior to 2007 may be outdated and other reviews have shown that production of publications involving e-health began to rise from 2007 onwards.[33,35]”

2. 'Study eligibility criteria': Please provide a rationale for why the excluded caregiver populations were excluded and define how psycho-education differs from education.

Response: Thank you for this feedback. The justification for excluding caregivers with severe mental health conditions is that the review is specifically focused on e-mental health interventions targeting mental health difficulties associated with the provision of informal care, for example depression, anxiety and stress, as opposed to targeting severe mental health difficulties. Caregivers of non-community dwelling care recipients provide care in a different setting and often spend less time providing care. While discussing your comments, we realized using the term palliative care did not align with the caregiver population we intended to exclude, as palliative care can be provided throughout the course of illness (Hawley, 2014; World Health Organization, 2020). We intended to exclude studies exclusively focused on caregivers of care recipients near the end-of-life (e.g. within a few months of death), as caregivers of individuals receiving end-of-life care have to cope with additional challenges such as grief and bereavement. We have replaced the term palliative with end-of-life, clarifying our understanding of this term and adding the justification for this decision. We have detailed these reasons further in the revised manuscript (see excerpt below). Generally, these exclusions were applied to improve the homogeneity of the review study populations, as the excluded caregiver populations experience unique challenges and caregiving situations.

Hawley, P. H. (2014). The Bow Tie Model of 21st Century Palliative Care. *Journal of Pain and Symptom Management*, 47(1), e2–e5. <https://doi.org/10.1016/j.jpainsymman.2013.09.007>

World Health Organization. (2020). WHO Definition of Palliative Care. <https://www.who.int/cancer/palliative/definition/en/>

Population section of the methods, page 5 line 16-26:

“Studies exclusively focusing on caregivers with severe mental health conditions (e.g. psychosis or bipolar disorder) will be excluded, as the focus of this review is on e-mental health interventions targeting psychological health difficulties associated with the provision of informal care, for example anxiety or depression, as opposed to targeting severe mental health conditions. Studies with interventions that solely focus on caregivers providing care to non-community dwelling care recipients will be excluded, given caregivers of individuals who do not live in the community may spend less time providing informal care[59] and generally experience lower levels of depression.[60,61] Additionally, studies of interventions designed specifically for caregivers of individuals at the end-of-life (e.g. within a few months of death) will be excluded, as end-of-life caregiving is associated with additional needs and burdens, for example difficulties related to grief and bereavement.[62]”

Based on a meta-analysis of psychoeducational interventions for depression, anxiety and psychological distress by Donker et al. 2009, we have now defined what psychoeducation can include on page 5 line 34-36:

“Psychoeducation is defined as the provision of information regarding common psychological health difficulties and can be delivered passively (e.g. an information website) or actively (e.g. an information website with therapist support, homework or exercises).[63]”

3. 'Outcomes': Please clarify if the reliability of outcome measures will be based on caregiver populations or a general population as outcome measures may perform differently in different populations.

Response: Thank you for this comment. After some discussion, we have decided to base the reliability of outcomes measures on the main validation paper of the outcome measure given studies in this review will likely include different caregiver populations, ages, genders and languages and the specific sample characteristics of an individual study are unlikely to match with those in validation papers. This statement has been added to the manuscript:

Outcomes section, page 6 line 16-19:

“Reliability of outcome measures will be assessed based on the main validation paper of the relevant measurement instrument, as this review will likely include studies with different caregiver populations, ages, genders and languages, the combination of which may not have been validated.”

4. 'Search Strategy': Please report what steps will be taken when identifying registered trial and other grey literature.

Response: Thank you for this comment, we have now elaborated on this in the revised manuscript, under “Study selection”.

Page 8 Line 19-25:

“Records retrieved from searches of clinical trial registries and OpenGrey will be screened for eligibility by one reviewer. When relevant clinical trial registries are identified, any resulting publications will be retrieved and screened for inclusion, unless already captured by the electronic database searches. If results from relevant trial registries are unpublished, authors will be contacted to determine if they are able to share details of any available results. Authors of grey literature records that do not contain enough information to assess eligibility will also be contacted for additional study details.”

Reviewer: 2

Reviewer Name

Miriam C. Noonan

Institution and Country

University of Exeter, United Kingdom

Please state any competing interests or state ‘None declared’:

None declared

Please leave your comments for the authors below I enjoyed reading this protocol and I thought the intro/background was very well written and presented the topic very well. It is an interesting topic and I look forward to reading the outcomes of the review.

I would just suggest some minor revisions.

1. Pragmatic RCT - please explain this much earlier in the text I found it confusing to grasp between RCTs for your QCA analysis and the fact that you are examining any study for your Thematic synthesis. Can this be made clearer earlier. e.g. you use the word reports at one point to describe the studies you would be looking at for your Thematic synthesis and this was confusing as I wasn't sure what type of reports this would be.

Response: Thank you for this comment. This is outlined in the "Study Designs" section but we agree, it would improve clarity to introduce information about the difference in study designs for each analysis method earlier. The "Comparators" section is the first section in which the PICOS criteria for each analysis type differs. We have therefore added information about study design in the "Comparators" section, leaving more extensive details (e.g. how we will assess if a trial is pragmatic) in the "Study Designs" section. Below is the revised section of the manuscript:

Page 6 Line 4-12:

"As it is necessary to determine effect sizes for the qualitative comparative analysis,[64] only studies of pragmatic randomized controlled trials with non-active controls will be included in this analysis. Non-active controls include: no treatment, wait-list control, treatment as usual, non-specific treatment component control (e.g. control for attention) or education on the care recipient's condition.[65] Studies using psychoeducation or active controls (e.g. controls using specific treatment components or studies comparing two therapies) will be excluded. For thematic synthesis of barriers and facilitators to implementation, studies of any design (e.g. randomized controlled trials, process evaluations, focus groups) will be included in the analysis, regardless of the presence or absence of a control."

2. Include rationale for why you are excluding studies with caregivers with mental health issues, non-community living patients and palliative care.

Response: Thank you for this comment. A more detailed justification of these exclusion criteria has been provided in the revised manuscript. A similar comment was made by Reviewer 1, so please see our response to Reviewer 1, comment #2. The revised section of the manuscript is as follows:

Population section of the methods, page 5 line 16-26:

"Studies exclusively focusing on caregivers with severe mental health conditions (e.g. psychosis or bipolar disorder) will be excluded, as the focus of this review is on e-mental health interventions targeting psychological health difficulties associated with the provision of informal care, for example anxiety or depression, as opposed to targeting severe mental health conditions. Studies with interventions that solely focus on caregivers providing care to non-community dwelling care recipients will be excluded, given caregivers of individuals who do not live in the community may spend less time providing informal care[59] and generally experience lower levels of depression.[60,61] Additionally, studies of interventions designed specifically for caregivers of individuals at the end-of-life (e.g. within a few months of death) will be excluded, as end-of-life caregiving is associated with additional needs and burdens, for example difficulties related to grief and bereavement.[62]"

3. Interventions (PICO) - Can you clarify whether RCTs, observational, cohort etc.

Response: We feel this information is best addressed in the "Study designs" section of the PICOS, leaving the "Intervention" section to detail the type of interventions eligible for inclusion. Also, as outlined in response to your first comment, we have added some of this information to the "Comparator" section as well to improve clarity for readers.

4. Thematic synthesis - needs a bit more clarity on how this will be conducted. It reads more that it is a framework analysis and you will be generating themes. More explanation on how you plan to integrate the data sets and how you will take into account Quant. elements that may be lost in synthesis e.g. sample size, attrition, duration of intervention etc.

Response: Thank you for this comment. We have tried to further clarify how the thematic synthesis will be conducted, particularly with respect to the last step of generating analytical themes and identifying barriers and facilitators to implementation. Although we use a primarily deductive coding approach, we are following the method outlined by Thomas and Harden, 2008 (referenced in the manuscript), which outlines how to use thematic synthesis for systematic reviews to identify barriers and facilitators. One aspect which we have elaborated on in the revised manuscript is that we will synthesize analytical themes related to, but going beyond, identified barriers and facilitators and the codes developed during the initial steps of the synthesis. We feel this differs from a framework analysis which may develop or modify a framework or model to explain a phenomenon (as in Carroll et al. 2013).

Carroll, C., Booth, A., Leaviss, J., & Rick, J. (2013). "Best fit" framework synthesis: Refining the method. *BMC Medical Research Methodology*, 13, 37. <https://doi.org/10.1186/1471-2288-13-37>

Thomas, J., & Harden, A. (2008). Methods for the thematic synthesis of qualitative research in systematic reviews. *BMC Medical Research Methodology*, 8, 45. <https://doi.org/10.1186/1471-2288-8-45>

Page 11 line 5-12:

"Results of the initial coding of qualitative data and narrative summaries of quantitative data will be analyzed together to identify barriers and facilitators to implementation. Two reviewers will independently identify barriers and facilitators, followed by discussion involving a third reviewer as needed.[93] Through this discussion, more abstract, analytical themes will be developed that go beyond the initial codes and identified barriers and facilitators.[93] This process will be iterative, modifying barriers and facilitators after defining initial analytical themes, followed by further refinement of analytical themes until the analytical themes fully encompass all codes and identified barriers and facilitators.[93]"

Descriptive data of studies included in the thematic synthesis will still be extracted and reported in the final review manuscript in the form of tables which summarize study (e.g. sample size, sample demographics, etc) and intervention (e.g. duration, number of sessions, level of support etc) characteristics. In this way, contextual information for each study will still be available to readers. This has been stated in the revised manuscript:

Page 9 line 16-18:

"Data related to the characteristics of each included study, such as the sample (e.g. sample size, participant demographics) or intervention (e.g. duration, type of support provided, delivery mode) characteristics, will be reported in summary tables."

Reviewer: 3

Reviewer Name

Heejung Kim

Institution and Country

Yonsei University, College of Nursing, Republic of Korea

Please state any competing interests or state 'None declared':

None declared

Please leave your comments for the authors below Thank you for giving me an opportunity to review this protocol. I expect that this paper will be very helpful for caregiving research and practitioners. However, there are few concern you need to consider. I hope that my review helps you.

1. This review will be very heterogeneous because it included many types of disease, both qualitative and quantitative methods, and diverse caregiver's psychological outcomes. So, it is very concerned that it will be difficult to synthesize the findings to draw key conclusion. Thus, you may think about narrowing down the scope of diseases and caregiver's outcomes.

Response: Thank you for your comment. We agree that the heterogeneity in psychological outcomes and care recipient health conditions will be a limitation of this review and we will acknowledge this in the results manuscript. However, we feel that taking a broader approach to the care recipient's health condition and caregiver psychological outcomes was needed. In the existing literature concerning e-mental health interventions for informal caregivers, interventions generally have broader targets concerning mental health rather than specifically targeting, for example, depression. Therefore we felt we needed to similarly take a broader approach to defining the mental health outcomes and intervention targets eligible for inclusion in this review.

In regards to the variety of care recipient's health conditions eligible for inclusion, caregivers of individuals with chronic diseases do experience common challenges related to providing different types of support and often experience common mental health difficulties (e.g., depression and anxiety) as a result of taking on a caregiving role. For example, one study comparing caregivers of people with cancer, dementia, diabetes or frail elderly, showed similarities between cancer and dementia caregivers related to types of care tasks, and caregiver physical strain and emotional stress.(Kim and Schulz, 2008) Additionally, as the present protocol paper is focused on barriers and facilitators to implementation, some barriers (e.g. usability) may be unrelated to the care recipient's disease, rather are more generally connected to intervention features or caregiver characteristics etc. Again, the heterogeneity in the studies we review and whether this influences implementation barriers and facilitators will be very important to consider in the context of the results of this review and we are grateful to you for highlighting this important point.

Kim, Y., & Schulz, R. (2008). Family caregivers' strains: Comparative analysis of cancer caregiving with dementia, diabetes, and frail elderly caregiving. *Journal of Aging and Health*, 20(5), 483–503. <https://doi.org/10.1177/0898264308317533>

2. In contrast, you set up too narrow scope for the eHealth. You seem to include only mobile- internet based. However, recently, there are many types along with smart devices. I suggest that you have consultation with ITC experts about this issue.

Response: Thank you for this feedback and we apologize for the lack of clarity in the manuscript. Mobile applications were only an example of a type of delivery mode that would be included. We have refined the phrasing to make sure it is clear that mobile applications are one of many types of devices that would be eligible for inclusion in the study.

Page 5 line 28-29:

“Interventions will utilise internet technology, such as web-based platforms or mobile-based applications, to deliver a mental health intervention to caregivers.[27,41]”

VERSION 2 – REVIEW

REVIEWER	Miriam Noonan College of Medicine and Health University of Exeter
REVIEW RETURNED	16-Apr-2020

GENERAL COMMENTS	Some minor comments: Strengths and limitations: The mixed method design of this review will ensure a wide variety of data on implementation is captured – make this stronger i.e. its not just a strength because it’s a wide variety of data. It enables a holistic interpretation of your findings Crisp set qualitative comparative analysis will make results of this review more concrete and usable for healthcare professionals and decision-makers – Crisp set qualitative comparative analysis facilitates understanding of the results increasing the usability of the findings for healthcare professionals and decision-makers Introduction: Additionally, although efficacy (also referred to as explanatory) trials are a useful tool to establish the beneficial effects of an intervention under ideal settings – I would suggest deleting this part of the sentence (lines 24-36) Commonly, systematic reviews and meta-analyses do not distinguish between pragmatic and explanatory trials despite the different conditions (e.g. setting, recruitment methods, eligibility criteria, control of adherence to and delivery of the intervention) under which interventions are evaluated. (delete sentence – lines 28-31) The majority of the intervention must be web-based – what kind of majority - -i.e. one supplemented telephone call or weekly telephone calls ok? Are you considering the inclusion of process evaluations for synthesis? - can this be specified if so – as this can enhance understanding of the implementation from RCT trials.
---

REVIEWER	Heejung Kim Yonsei University, Colleg of Nursing and Mo-Im Kim Nursing Research Institute, Republic of Korea
REVIEW RETURNED	19-Apr-2020

GENERAL COMMENTS	Thank you for your careful revision. I confirmed that the revision is clearly done abased on the previous review.
---

VERSION 2 – AUTHOR RESPONSE

Reviewer(s)' Comments to Author:

Reviewer: 2

Reviewer Name

Miriam Noonan

Institution and Country

College of Medicine and Health

University of Exeter

Please state any competing interests or state 'None declared':

None declared

Please leave your comments for the authors below Some minor comments:

Strengths and limitations:

1. The mixed method design of this review will ensure a wide variety of data on implementation is captured – make this stronger i.e. its not just a strength because it's a wide variety of data. It enables a holistic interpretation of your findings

Thank you for this comment. We agree and have added to this strength to highlight that the mixed method design also enhances our interpretations:

“The mixed method design of this review will ensure a wide variety of data on implementation is captured and interpretations account for both qualitative and quantitative research findings”

2. Crisp set qualitative comparative analysis will make results of this review more concrete and usable for healthcare professionals and decision-makers – Crisp set qualitative comparative analysis facilitates understanding of the results increasing the usability of the findings for healthcare professionals and decision-makers

Thank you for suggestions on the phrasing of the strengths and limitations. This strength has been rephrased:

“Crisp set qualitative comparative analysis produces concrete results, increasing the usability of findings for healthcare professionals and decision-makers”

Introduction:

1. Additionally, although efficacy (also referred to as explanatory) trials are a useful tool to establish the beneficial effects of an intervention under ideal settings – I would suggest deleting this part of the sentence (lines 24-36)

This has been removed.

2. Commonly, systematic reviews and meta-analyses do not distinguish between pragmatic and explanatory trials despite the different conditions (e.g. setting, recruitment methods, eligibility criteria, control of adherence to and delivery of the intervention) under which interventions are evaluated. (delete sentence – lines 28-31)

We feel this statement adds value by highlighting that reviews do not often differentiate between studies that have pragmatic designs, which has in part motivated the methods we have selected to differentiate between pragmatic and explanatory trials. We have rephrased the sentence and added a supporting statement to strengthen the importance of considering trial design in reviews (pg 4, line 27-32):

“However, systematic reviews and meta-analyses do not often distinguish between pragmatic and explanatory (also referred to as efficacy) trials despite the different conditions (e.g. setting, recruitment methods, eligibility criteria, control of adherence to and delivery of the intervention) under which interventions are evaluated.[50,52] Identifying trials with a pragmatic design may be a valuable factor to consider when interpreting results of reviews to inform implementation.”

3. The majority of the intervention must be web-based – what kind of majority - -i.e. one supplemented telephone call or weekly telephone calls ok?

We agree that this statement could be clarified in the manuscript and we thank you for highlighting the lack of clarity here. We require that the therapeutic materials are provided online, with little restrictions on the type and/or amount of additional support provided. This has been clarified in the text (pg 5, line 36-39):

“The majority of therapeutic materials within the e-mental health intervention must be internet based, however, this may be supplemented with additional forms of support (such as telephone contact, face-to-face support or video-conferencing). There are no restrictions on the amount of support provided within the e-mental health intervention.”

4. Are you considering the inclusion of process evaluations for synthesis? - can this be specified if so – as this can enhance understanding of the implementation from RCT trials.

This has been mentioned in the methods section “Comparators” (pg 6, line 10-12):

“For thematic synthesis of barriers and facilitators to implementation, studies of any design (e.g. randomized controlled trials, process evaluations, focus groups) will be included in the analysis, regardless of the presence or absence of a control.”

Reviewer: 3

Reviewer Name

Heejung Kim

Institution and Country

Yonsei University, Colleg of Nursing and Mo-Im Kim Nursing Research Institute, Republic of Korea

Please state any competing interests or state ‘None declared’:

None declared

Please leave your comments for the authors below Thank you for your careful revision. I confirmed that the revision is clearly done abased on the previous review.